# Comparison of the Efficacy and Safety of Biologics (Secukinumab, Ustekinumab, and Guselkumab) for the Treatment of Moderate-to-Severe Psoriasis: Real-World Data from a Single Korean Center

**DOI:** 10.3390/biomedicines10051058

**Published:** 2022-05-03

**Authors:** Seung-Won Jung, Sung Ha Lim, Jae Joon Jeon, Yeon-Woo Heo, Mi Soo Choi, Seung-Phil Hong

**Affiliations:** 1Department of Dermatology, Yonsei University Wonju College of Medicine, Wonju 26426, Korea; seungwon0826@naver.com (S.-W.J.); limsh7812@naver.com (S.H.L.); wowns_1228@naver.com (J.J.J.); hywoo619@naver.com (Y.-W.H.); 2Department of Dermatology, Dankook University College of Medicine, Cheonan 31116, Korea; misu2532@naver.com

**Keywords:** psoriasis, direct comparison in real world, secukinumab, ustekinumab, guselkumab

## Abstract

Biologics are important treatment options for psoriasis; however, direct comparison of their efficacy, safety, and drug survival is insufficient in clinical practice. This retrospective single-center study aimed to compare the efficacy, safety, and drug survival of three commonly used psoriasis biologics (secukinumab, ustekinumab, and guselkumab) and identify the factors affecting drug survival in actual clinics in Korea. We enrolled 111 patients with moderate to severe psoriasis and for at least 56 weeks of follow-up; among these, 27, 23, and 61 were administered secukinumab, ustekinumab, and guselkumab, respectively. All groups were comparable with respect to their baseline characteristics. Secukinumab showed a rapid response, and guselkumab was superior in terms of a long-term response and complete remission compared with other biologics, while ustekinumab showed a lower efficacy compared with other biologics. All three biologics had a favorable and similar safety profile; however, allergic reactions and latent tuberculosis were more common with secukinumab and ustekinumab, respectively. Guselkumab was the most sustained biologic, and the survival rates of secukinumab and ustekinumab were similar. Drug survival was remarkably shorter in female patients and those with hypertension. Introduction of new biologics emerged as a negative factor for drug survival in clinical settings.

## 1. Introduction

Psoriasis is a chronic inflammatory skin condition characterized by scaly erythematous patches or plaques affecting prominent extensor surfaces but spreading to whole body areas, including flexor surfaces [1]. There are various clinical phenotypes of psoriasis, such as palmoplantar, inverse, guttate, and pustular [2]. Psoriasis occurs when the immune system attacks the skin; the interleukin (IL)-12 and IL-17/23 axis plays major roles in its pathogenesis. Prevalence varies depending on geography, ethnicity, and genetic and environmental factors, and the annual standardized prevalence in Korea is estimated to be 0.45% according to recent studies [3,4,5]. Psoriasis profoundly impairs patients’ health-related quality of life (QOL), as psoriasis patients report a reduction in physical and mental functioning comparable to that observed with other major medical diseases, such as cancer, hypertension, heart disease, diabetes, and depression [6]. However, biologics targeting IL-17 or IL-23 have emerged as an important treatment option for psoriasis and have led to a substantial improvement in the QOL of patients owing to their superior effect on decreasing subjective disease burden compared with conventional therapies [7].

Various biologics are used for the treatment of psoriasis in clinical practice; these include secukinumab (a fully humanized anti-IL-17A IgG1κ monoclonal antibody), ixekizumab (a humanized anti-IL-17A IgG4 monoclonal antibody), ustekinumab (a humanized anti-IL-12/23 p40 subunit IgG1κ monoclonal antibody), guselkumab (a fully humanized anti-IL-23 p19 subunit IgG1 monoclonal antibody), and risankizumab (a fully humanized anti-IL-23 p19 subunit IgG1λ monoclonal antibody). Multiple previous psoriasis studies evaluating the relative efficacy of biologics have concluded that biologics targeting IL-17A, such as secukinumab and ixekizumab, show superior efficacy compared with IL-12/23 inhibitors, such as ustekinumab; however, safety profiles of both groups have been found to be similar [8,9]. In addition, improved efficacy profiles of IL-23 p19 inhibitors versus IL-12/23 p40 inhibitors have been confirmed with comparable safety outcomes [10]. However, these clinical trials of biologics primarily focused on rapid responses (until weeks 12–16). As psoriasis is a chronic and incurable disease, long-term responses (approximately 1 year) and complete remission are more relevant (compared with short-term responses) in real clinical practice. In addition, the results of randomized controlled clinical trials may not match the observations made during real-world clinical practice as patients in clinics may have multiple variables (comorbidities, polypharmacy, etc.) and are more complicated than study subjects. In Korea, direct comparisons of the efficacy and safety of psoriasis biologics are insufficient in clinical practice. Furthermore, little is known about the drug survival rates of psoriasis biologics and the factors affecting these rates.

Therefore, we aimed to compare the efficacy and safety of the three commonly used biologics used in the treatment of psoriasis (secukinumab, ustekinumab, and guselkumab) in actual clinics of Korea. Additionally, we aimed to identify the factors that independently affect the survival rates of these drugs.

## 2. Materials and Methods

### 2.1. Study Design and Patients

This was an observational retrospective single-center cohort study that analyzed clinical data extracted from patients with psoriasis registered at Wonju Severance Christian Hospital from August 2012 to July 2021. Eligible patients were 18 years old and had moderate to severe psoriasis (psoriasis area and severity index (PASI) ≥ 10 and body surface area involvement (BSA) ≥ 10%) recalcitrant to nonbiologic systemic immunosuppressive treatments (cyclosporine or methotrexate) or phototherapy (narrowband ultraviolet B) for more than 6 months. All included patients were naïve to biologic therapy that they had never been treated with biologic therapy, and stopped previous systemic treatments and phototherapy after initial biologic injection. Only patients treated with at least one injection of secukinumab, ustekinumab, or guselkumab with at least 56 weeks of follow-up were included in the study. This study was approved by the Yonsei University Wonju Campus Institutional Review Board and was performed in accordance with the relevant guidelines (No. CR321184, 22 February 2022).

### 2.2. Procedure

Patients were administered with either secukinumab, ustekinumab, or guselkumab, with a follow-up of at least 56 weeks. Secukinumab 300 mg (Cosentyx; Novartis Pharmaceuticals Corporation, East Hanover, NJ, USA) was injected as two 150 mg subcutaneous injections at weeks 0, 1, 2, 3, and 4 and every 4 weeks thereafter. Ustekinumab (Stelara; Janssen Research and Development, Spring House, PA, USA) was administered at a dose of 45 mg initially and after a period of 4 weeks, followed by a dose of 45 mg administered every 12 weeks. Guselkumab 100 mg (TREMFYA; Janssen Research and Development, Spring House, PA, USA) was injected subcutaneously at weeks 0, 4, and 12 and every 8 weeks thereafter.

### 2.3. Outcomes

The primary outcomes of this study were efficacy profiles such as proportion of patients in each treatment group achieving PASI 75 (i.e., ≥75% improvement in PASI) or 90 at week 16 for rapid response, PASI 90 at week 56 for long-term response, and PASI 100 at week 56 for complete remission. Adverse events (AEs) were defined as the observation of at least one adverse event following the injection of the respective biologic and serious adverse events (SAEs) were defined as AEs resulting in the discontinuation of the biologic injection or changing to other biologics. Drug survival rates of the biologics were calculated as the period from the first biologic injection to biologic discontinuation or alteration (prescribing a different biologic) due to loss of efficacy or SAEs.

### 2.4. Statistical Analysis

Categorical variables, presented as the number and proportion of patients, were compared using the Pearson chi-square test. Continuous variables are presented as mean ± standard deviation and were analyzed using two-way analysis of variance (ANOVA). We used the Kaplan–Meier method and log-rank test to analyze drug survival rates and the Cox proportional hazard regression analysis to identify demographic factors affecting drug survival. Statistical significance was set at *p* < 0.05.

## 3. Results

### 3.1. Baseline Characteristics

A total of 111 patients were enrolled in this analysis, with 27 (24.3%), 61 (55.0%), and 23 (20.7%) receiving secukinumab, ustekinumab, and guselkumab, respectively. Patients treated with ixekizumab and risankizumab were excluded from this study as the number of those treated with ixekizumab was small (5) and the follow-up period for risankizumab was less than 1 year. Basic demographic features were comparable among the patients in the three treatment groups, and psoriasis was more common in male patients than in female patients (Table 1). All included patients were naïve to biologic therapy and had been treated with systemic immunosuppressive treatments (cyclosporine or methotrexate) or phototherapy for more than 6 months. Initial PASI and comorbidities were comparable among the three biologic groups, while diabetes seemed to be common in the secukinumab group, and dyslipidemia seemed to be uncommon in the ustekinumab group, but they were not statistically significant. Hypertension (24.3%) was the most common comorbidity, followed by dyslipidemia (18.9%) and diabetes (8.1%). Of the patients treated with biologics, 8.1% had psoriatic arthritis (PsA).

### 3.2. Direct Comparison of the Efficacy of the Biologics in Real-World Practice

With respect to the PASI 75 response, secukinumab was significantly superior to guselkumab (88.9% for secukinumab vs. 55.2% for guselkumab, *p* < 0.01) and ustekinumab (88.9% vs. 33.4% for secukinumab, *p* < 0.001) in the early phase, such as week 16 (Figure 1a). Until week 56, ustekinumab showed lower efficacy with respect to the PASI 75 response, and the superior efficacy of secukinumab was replaced by that of guselkumab in the latter phase of the treatment. Further, differences between the efficacies of the three biologics with respect to the PASI 90 response were similar to those observed with the PASI 75 response (Figure 1b). With respect to the PASI 90 response at week 16, secukinumab showed the greatest efficacy compared with guselkumab (74.1% for secukinumab vs. 39.1% for guselkumab, *p* = 0.013) and ustekinumab (74.1% for secukinumab vs. 9.8% for ustekinumab, *p* < 0.001), with ustekinumab showing the lowest efficacy among the three biologics (39.1% for guselkumab vs. 9.8% for ustekinumab, *p* = 0.002). In the latter phase (week 56), guselkumab showed greater efficacy than secukinumab; however, the difference was not significant (91.3% for guselkumab vs. 81.5% for secukinumab, *p* = 0.318). In particular, guselkumab showed a substantially greater efficacy than ustekinumab (91.3% vs. 65.6%, *p* = 0.018), and so did secukinumab (81.5% vs. 65.6%, *p* = 0.131). At week 56, a significantly lower proportion of patients in the ustekinumab group achieved PASI 100 compared with those in the guselkumab group (82.6% for guselkumab vs. 36.1% for ustekinumab, *p* < 0.001) and secukinumab group (63.0% for secukinumab vs. 36.1% for ustekinumab, *p* = 0.019) (Figure 1c). Similar to the PASI 75 and 90 responses, a higher proportion of patients in the guselkumab group achieved a PASI 100 response compared with those in the secukinumab group; however, the difference was not significant (82.6% for guselkumab vs. 63.0% for secukinumab, *p* = 0.123).

### 3.3. Direct Comparisons of Biologic Safety in Real-World Practice

Similar proportions of AEs among biologics were found, with 18.5% (5/27), 18.0% (11/61), and 17.4% (4/23) patients experiencing at least one AE with secukinumab, ustekinumab, and guselkumab, respectively (Table 2). Out of the 61 patients receiving ustekinumab, 5 (8.2%) with no history of tuberculosis (TB) were diagnosed as having latent TB using the QuantiFERON-TB Gold test after ustekinumab injection. Patients taking ustekinumab had the highest proportion of fungal infections (6.6%; 4/61), and all of them had mild tinea corporis without candidiasis. In contrast, no fungal infections, including candidiasis, were identified in the secukinumab group. Allergic reactions were mainly identified in the secukinumab group; local injection site reactions were observed in 7.4% (2/27) and systemic allergic reactions in 3.7% (1/27) of these patients. The AEs resulting in biologic discontinuation or change were defined as severe adverse events (SAEs). Only 4 patients with SAEs (3.6%) were reported: 3 in the secukinumab group (2 due to injection site reactions resulting in biologic discontinuation, and 1 due to systemic allergic reaction resulting in biologic changes to guselkumab) and 1 in the ustekinumab group (due to worsening of psoriasis, resulting in biologic changes to risankizumab). In the secukinumab group, SAEs were more common than the other two biologics, and all of them were related to local or systemic allergic reactions.

### 3.4. Direct Comparisons of Biologic Drug Survival up to 56 Weeks in Real-World Practice

Guselkumab showed the longest drug survival of up to 56 weeks. The overall drug survival rate at week 56 was 95.7% (22/23) in patients treated with guselkumab, 81.5% (22/27) for secukinumab, and 73.8% (45/61) for ustekinumab (Figure 2). Significant differences were observed in the drug survival rates of guselkumab and ustekinumab at week 56 (*p* = 0.003), without any significant differences between other attributes of these two biologics. Although ustekinumab showed longer drug survival than secukinumab did till 52 weeks, this effect was reversed at week 52 onwards. Out of 27 patients treated with secukinumab, 5 patients were lost during follow-up period up to 56 weeks; 2 patients discontinued the injection due to SAE (2 patients due to injection site reactions), and 3 patients changed biologic due to SAE (1 patient due to systemic allergic reaction) and loss of efficacy (2 patients) (Table 3). Out of 61 patients, 16 patients treated with ustekinumab were lost during follow-up period up to 56 weeks; 3 patients discontinued the treatment due to SAE (worsening of psoriasis), loss of efficacy, and complete remission willing to self-discontinue, respectively. The other 13 patients lost during follow-up changed biologics due to loss of efficacy. Loss of efficacy resulting in biologic change was more common for ustekinumab, and most of them occurred around a year (52–56 weeks) after biologic initiation. Only 1 patient out of 23 patients treated with guselkumab changed biologics due to loss of efficacy, and there were no patients of biologic discontinuation.

### 3.5. Factors Affecting Drug Survival in Real-World Practice

The factors affecting drug survival based on the patient baseline characteristics were analyzed. We found that most of the factors were not significantly associated with drug survival. However, female patients showed a shorter drug survival rate than male patients (74.4% and 83.3%, respectively) up to 56 weeks, and female sex was significantly associated with a higher biologic discontinuation or change (hazard ratio (HR) (95% CI): 1.38 (1.12–1.68), *p* = 0.040) (Table 4). Although it was not statistically significant, there was a tendency that the higher the initial PASI score, the less biologic discontinuation or change. Alcohol consumption was a positive predictor for biologic discontinuation or change, whereas current smoking was a negative predictor; however, the results were not significant. Among the various comorbidities, only hypertension significantly increased the risk of drug survival shortening (84.5% vs. 66.7% for patients without hypertension; HR (95% CI): 1.36 (1.13–1.57), *p* = 0.020). Patients with other comorbidities, such as diabetes, hepatitis, tuberculosis, and obesity (BMI ≥ 30), showed a higher tendency for biologic discontinuation or change; however, these differences were not significant. Among the various comorbidities, dyslipidemia and psoriatic arthritis were predictors of drug survival, although the differences were not significant. Biologic change from ustekinumab or secukinumab to guselkumab significantly increased after the initiation of guselkumab irrespective of the treatment that had been injected, and drug survival was significantly decreased after the introduction of guselkumab (HR (95% CI): 3.42 (2.81–4.17), *p* = 0.002).

## 4. Discussion

IL-23 induces the production of proinflammatory mediators, such as IL-17A and IL-17F, through Th17 cell activation. This IL-23/Th17 axis plays a major role in psoriasis pathogenesis [11,12]. However, there are limited reports on IL-23 inhibitors concerning their efficacy and safety in real-world settings given their recent introduction in the market. Real-world data are needed to identify promising treatment options for patients with psoriasis. The present study is of particular importance because, in Korea, real-world data of other biologics and IL-23 inhibitors are scarce.

We found that patients administered secukinumab showed a rapid response, while guselkumab was superior in terms of long-term response (approximately 1 year) and complete remission compared with other biologics. Among all the assessed biologics, ustekinumab demonstrated a relatively low efficacy. Patients treated with secukinumab demonstrated a rapid response to initial sufficient booster injections (administered as two 150 mg subcutaneous injections at weeks 0, 1, 2, 3, and 4 every 4 weeks). Biologics related to the IL-23/Th17 axis, such as those selectively targeting IL-23 p19 or IL-17A, are superior to biologics targeting IL-12/23 p40 [9,10,13]. We also observed that ustekinumab showed the lowest efficacy among all efficacy profiles in the enrolled Korean clinics. Recently, selective IL-23 inhibitors, such as guselkumab and risankizumab, have been reported to show promising efficacy compared with drugs targeting other inflammatory mediators [10,14,15,16,17,18]. Especially, the superiority of guselkumab in terms of efficacy profiles was recently identified in Asians, including Koreans [19,20]. Although none of these studies used long-term clinical data, the superiority of guselkumab and inferiority of ustekinumab over 56 weeks were similarly identified in our real-world data analysis. As such, the biologic agent itself can have a significant impact on efficacy; however, a history of previous biological treatments can also negatively affect efficacy and drug survival according to recent studies [21,22].

All three biologics had favorable and similar safety profiles; however, severe allergic reactions, resulting in biologic discontinuation or change, were significantly more common in the secukinumab group. Three patients treated with secukinumab discontinued or changed the treatment regimen because of local or systemic allergic reactions. These cases were considered to have had delayed-type drug sensitivity reactions, as they occurred after multiple injections. We discontinued the secukinumab injection immediately; two local allergic patients improved with topical corticosteroid treatment, and one systemic allergic reaction patient changed to guselkumab without any AEs. As more number of patients in the secukinumab group exhibited severe allergic reactions than those treated with other biologics, we suggest that it is better to inject other biologics if there is a history of allergy or adverse drug reactions. In addition, clinicians should be more concerned with patients given secukinumab than those given other biologics to identify allergic reactions. Among those treated with ustekinumab, latent TB was identified in five patients without a history of TB. We temporarily discontinued ustekinumab administration and resumed it after completing an anti-TB medication for several months. In a recent Taiwanese study, the seroconversion rate (assessed using the QuantiFERON-TB Gold test) following ustekinumab administration was reported to be low (7.3%), with no active TB being detected [23]. The prevalence of TB in Korea is 8.2% higher than that reported in this Taiwanese study. We found that patients treated with ustekinumab had common fungal infections; none of them were identified in patients treated with secukinumab. As IL-17-mediated immune responses can target *Candida albicans*, using IL-17 inhibitors can increase the risk of *Candida* infections [24]. Unlike previous studies, *Candida* infections were uncommon in patients treated with the three biologics. Among those experiencing SAEs, patients with worsening psoriasis taking secukinumab and ustekinumab were switched to guselkumab or risankizumab. Other common AEs of psoriasis biologics, such as nasopharyngitis, upper respiratory tract infection, headache, arthralgia, diarrhea, and conjunctivitis, were rarely identified. This may explain why these patients did not recognize it as an adverse reaction during treatment, because these events were mild and transient.

According to the results of a Danish DERMBIO registry study comparing the drug survival of adalimumab, etanercept, infliximab, secukinumab, and ustekinumab, ustekinumab was associated with the highest drug survival, while secukinumab was associated with the lowest drug survival. [25]. However, real-world data from BADBIR showed similar drug survival functions for the two biologics at both the first year (0.88 for secukinumab (95% CI; 0.86−0.91) vs. 0.88 for ustekinumab (95% CI; 0.87−0.89)) and the second year (0.77 for secukinumab (95% CI; 0.73−0.80) vs. 0.77 for ustekinumab (0.76−0.79)), and little is known about the drug survival rate of guselkumab owing to its more recent approval and availability [26]. In our clinical study, guselkumab was the most sustained biologic throughout the 56 weeks of follow-up. In particular, ustekinumab had a higher drug survival rate than secukinumab until less than 1 year of treatment, with this statistic being reversed after 1 year of treatment. Compared with the other biologics, loss of efficacy was more commonly observed with ustekinumab; in a majority of such cases, ustekinumab therapy was replaced with an IL-23 inhibitor therapy at the 1 year follow-up. This observation suggests that the drug survival of ustekinumab continuously declined over time until 56 weeks, mainly because of the loss of efficacy. In addition, a dramatic decrease in drug survival around 1 year indicates that replacing ustekinumab with a different biologic for therapy was usually considered during the 1-year follow-up.

A recent study identified that female sex and certain comorbidities, such as hypertension and diabetes, are predictive of the poor persistence of psoriasis biologics, whereas PsA is predictive of good persistence [27]. In particular, female sex has been identified as an independent risk factor for biologic discontinuation, according to previous studies [28,29,30,31,32,33,34]. Our clinical data also showed a loss of adherence to biologic treatment in female patients; however, a direct cause for the same, such as loss of efficacy or AEs, was not identified. Previous studies have shown that metabolic syndrome (MS) comorbidities (hypertension, diabetes, dyslipidemia, and obesity (BMI ≥ 30)) affect drug survival [29,30,33,34,35]. In our study, the proportion of psoriasis patients with one or more metabolic conditions was 44.1% (49/111). In particular, psoriasis patients with MS comorbidities showed more drug discontinuation or drug change compared with patients without MS comorbidities (drug survival rates up to 56 weeks: 77.6% for patients with MS comorbidities vs. 82.3% for patients without MS comorbidities, *p* = 0.537). Further analysis revealed that patients with obesity or diabetes showed more drug discontinuation or drug change compared with patients without obesity or diabetes; however, the differences were not significant. Only hypertension significantly decreased drug survival out of the MS comorbidities in our study. MS comorbidities have emerged as a positive factor for drug survival due to the generally increased health awareness of patients with MS comorbidities, causing these patients to show a better tendency towards therapy adherence, which leads to increased drug persistence [33]. However, in real-world clinical settings, most MS comorbidities, especially hypertension, decrease drug survival. Therefore, clinicians must investigate the medical histories of patients with psoriasis and consider the possibility of biologic therapy noncompliance in patients with MS comorbidities. Additionally, the higher the initial PASI score seems the less biologic discontinuation or change to another biologics, which is assumed to be due to the tendency of severe patients to be treated steadily. In our clinical setting, PsA was predictive of good biological persistence and has been observed in a previous study; however, this result was not statistically significant. In particular, biologic change increased significantly after March 2020, when guselkumab was introduced in our clinic. Recently, the introduction of new biologics has emerged as a factor that reduces drug survival of the ongoing therapeutics [28]. The results of our study were similar to those of previous studies, and the availability of new or alternative biologics is considered an essential factor while calculating drug survival.

The limitations of our study included a small sample size of patients, a relatively short follow-up period of 1 year, and its monocentric and retrospective nature. An important limitation of this study was the exclusion of two other newest biologics approved in Korea, namely, ixekizumab and risankizumab. Ixekizumab was excluded due to its administration to a considerably small number of patients at our institution, and risankizumab was excluded because there were no sufficient long-term follow-up results (follow-up of at least 1 year) as it was recently introduced. In the future, we will analyze the real-world data on risankizumab and guselkumab treatments to analyze their relative efficacy, safety, and drug survival.

## 5. Conclusions

In conclusion, in Korean clinical practice, secukinumab showed a rapid effect, while guselkumab was more advantageous in terms of its long-term effects, potential for inducing complete disease remission, and drug survival rate. All three commonly used biologics had favorable and similar safety profiles. Female sex, hypertension, and availability of new biologics were determined to be negative factors for drug survival in a real-world clinical setting.

## Figures and Tables

**Figure 1 biomedicines-10-01058-f001:**
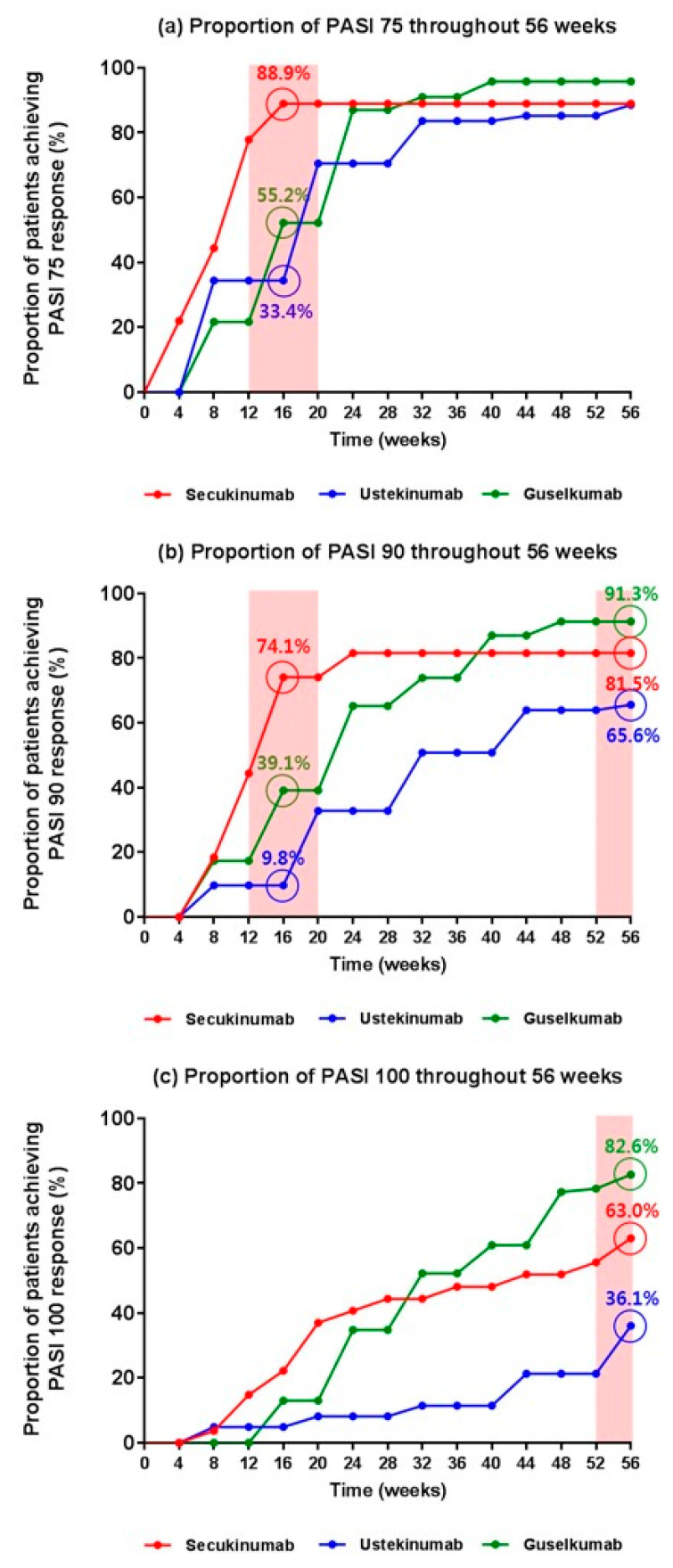
Proportion of patients achieving clinical response over time with secukinumab, ustekinumab, and guselkumab. The proportion of patients achieving > 75% (**a**), 90% (**b**), and 100% (**c**) improvement in PASI through week 56. Abbreviation: PASI, psoriasis area and severity index.

**Figure 2 biomedicines-10-01058-f002:**
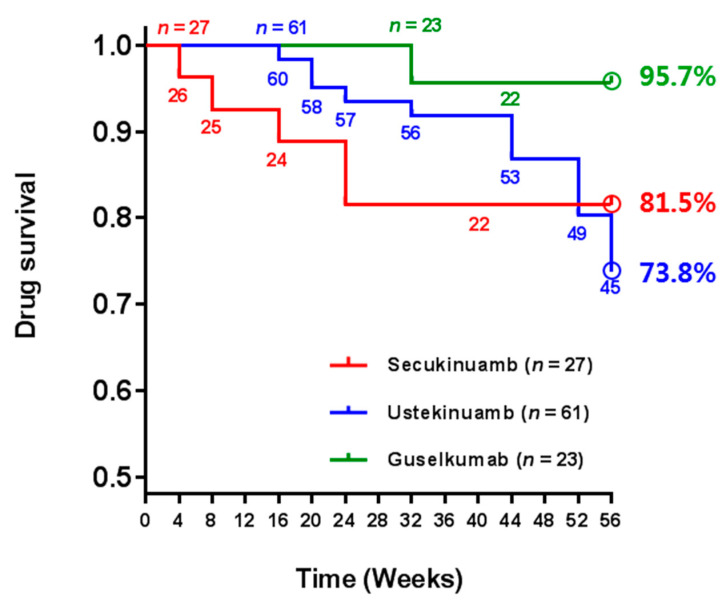
Drug survival rates of up to 56 weeks. The numbers of patients still receiving treatment for each biologic are described below the line in the figure.

**Table 1 biomedicines-10-01058-t001:** Baseline characteristics.

	Secukinumab (*n* = 27)	Ustekinumab (*n* = 61)	Guselkumab (*n* = 23)	*p*-Value
Age (years) Mean (SD)	47.9 (15.6)	47.8 (15.3)	43.6 (15.1)	0.82
Sex (female)	8 (29.6%)	23 (37.7%)	8 (34.8%)	0.76
Body mass index, kg/m^2^ Mean (SD) Obesity (BMI ≥ 30)	26.0 (4.9) 4 (14.8%)	24.6 (4.1) 9 (14.8%)	26.0 (5.1) 3 (13.0%)	0.82 0.99
Body surface area (%) Mean (SD)	25.0 (8.2)	26.8 (11.4)	27.4 (11.0)	0.27
Initial PASI Mean (SD)	17.9 (6.2)	17.5 (7.5)	20.0 (9.2)	0.14
Previous treatments Only systemic immunosuppressive Treatments ^†^ Sequential combination of immunosuppressive treatments and phototherapy ^††^ Alcohol	12 (44.4%) 15 (55.6%) 14 (51.9%)	30 (49.2%) 31 (50.8%) 33 (54.1%)	11 (47.8%) 12 (52.2%) 15 (65.2%)	0.92 0.92 0.59
Smoking	9 (33.3%)	29 (47.5%)	11 (47.8%)	0.43
Hypertension	8 (29.6%)	12 (19.7%)	7 (30.4%)	0.45
Diabetes	5 (18.5%)	3 (4.9%)	1 (4.3%)	0.07
Previous hepatitis history	1 (3.7%)	2 (3.3%)	3 (13.0%)	0.19
Previous tuberculosis history	0 (0.0%)	6 (9.8%)	2 (8.7%)	0.25
Dyslipidemia Cardiovascular disease Psoriatic arthritis	7 (25.9%) 1 (3.7%) 2 (7.4%)	7 (11.5%) 2 (3.3%) 4 (6.6%)	7 (30.4%) 1 (4.3%) 3 (13.0%)	0.08 0.97 0.62

Abbreviations: SD, standard deviation; IQR, interquartile range; BMI, body mass index; PASI, psoriasis area and severity index. ^†^ Systemic immunosuppressive treatments were methotrexate or cyclosporine. ^††^ Phototherapy was NB-UVB.

**Table 2 biomedicines-10-01058-t002:** Safety profiles.

	Secukinumab (*n* = 27)	Ustekinumab (*n* = 61)	Guselkumab (*n* = 23)	*p*-Value
Patients with ≥1 AE Tuberculosis Viral infection Fungal infection Injection site reaction Worsening of psoriasis Conjunctivitis Systemic allergic reaction Arthralgia	5 (18.5%) 0 (0.0%) 1 (3.7%) 0 (0.0%) 2 (7.4%) 0 (0.0%) 1 (3.7%) 1 (3.7%) 0 (0.0%)	11 (18.0%) 5 (8.2%) 0 (0.0%) 4 (6.6%) 1 (1.6%) 1 (1.6%) 0 (0.0%) 0 (0.0%) 0 (0.0%)	4 (17.4%) 0 (0.0%) 0 (0.0%) 1 (4.3%) 1 (4.3%) 0 (0.0%) 1 (4.3%) 0 (0.0%) 1 (4.3%)	0.99 0.07 0.22 0.39 0.42 0.66 0.30 0.22 0.15
SAE affecting drug survivals	3 (11.1%)	1 (1.6%)	0 (0.0%)	<0.01 **

** *p* < 0.01. Abbreviations: AE, adverse event; SAE, severe adverse event.

**Table 3 biomedicines-10-01058-t003:** Drug survival for up to 56 weeks.

	Secukinumab (*n* = 27)	Ustekinumab (*n* = 61)	Guselkumab (*n* = 23)
Drug survival for up to 56 weeks Median drug survival weeks (95% CI) Patients lost during follow-up period Biologic discontinuation Severe adverse event Loss of efficacy Complete remission	22/27 (81.5%) 48.4 (41.9–55.0) 5/27 (18.5%) 2 (7.4%) 2 (7.4%) 0 (0.0%) 0 (0.0%)	45/61 (73.8%) 52.4 (50.0–54.8) 16/61 (26.2%) 3 (4.9%) 1 (1.6%) 1 (1.6%) 1 (1.6%)	22/23 (95.7%) 55.0 (52.8–57.1) 1/23 (4.3%) 0 (0.0%) 0 (0.0%) 0 (0.0%) 0 (0.0%)
Biologic change Severe adverse event Loss of efficacy	3 (11.1%) 1 (3.7%) 2 (7.4%)	13 (21.3%) 0 (0.0%) 13 (21.3%)	1 (4.3%) 0 (0.0%) 1 (4.3%)

Abbreviations: CI, confidence interval.

**Table 4 biomedicines-10-01058-t004:** Factors inducing biologic discontinuation or change.

	HR (95% CI)	*p*-Value
Female sex Obesity (BMI ≥ 30) Initial PASI	1.38 (1.12–1.68) 1.59 (0.77–3.26) 0.91 (0.82–1.02)	0.04 * 0.36 0.09
Alcohol consumption	1.99 (0.45–8.85)	0.36
Current smoker	0.67 (0.35–1.28)	0.22
Hypertension	1.36 (1.13–1.57)	0.02 *
Diabetes	1.84 (0.64–5.32)	0.26
Hepatitis	1.18 (0.36–3.88)	0.79
Tuberculosis	1.45 (0.57–4.00)	0.47
Dyslipidemia	0.75 (0.31–1.83)	0.53
Psoriatic arthritis	0.88 (0.32–2.42)	0.81
After guselkumab introduction	3.42 (2.81–4.17)	<0.01 **

* *p* < 0.05, ** *p* < 0.01. Abbreviations: HR, hazard ratio; CI, confidence interval; BMI, body mass index; PASI, psoriasis area and severity index.

## Data Availability

Not applicable.

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
