# Peer review of "Comparison of the Efficacy and Safety of Biologics (Secukinumab, Ustekinumab, and Guselkumab) for the Treatment of Moderate-to-Severe Psoriasis: Real-World Data from a Single Korean Center"

_biomedicines, 2022, doi:10.3390/biomedicines10051058_

Round 1
Reviewer 1 Report
Comments for Authors:
This real-world retrospective study performed in Korea compared the efficacy and safety of secukinumab, ustekinumab and guselkumab. There are some comments that I have made that I wish the Authors to address.
- Figure 2: In KM drug survival figure, the number of patients still treated (for each biologic) should be shown below the graph.
- Table 1: p-values need to be added to show that there are no statistically significant differences between patients treated with the 3 different biologics. If differences emerge, these then need to be discussed.
- Table 1: previous biologic and naïve patients are missing. Other systemic treatment for psoriasis should also be shown. Current concomitant medication for comorbid diseases should be presented too.
- Table 1: only mean values are necessary to present, not median too.
- Table 1: did any of the patients have cardiovascular disease (considering the frequency of alcohol consumption and cigarette smokers + hypertension).
- Efficacy data should also take into account previous biological treatment.
- Obesity was defined as ≥ BMI of 25 kg/m2. This should be modified to “over-weight” as obesity is ≥ BMI of 30 kg/m2. However, the cox regression model should consider the predictor of obesity as BMI of ≥ BMI of 30 kg/m2 as predictor impacting on drug survival in their model.
- Why was baseline PASI not included as a predictor of drug survival in their model (Table 4)?
- Discussion: two other recent studies by Youn et al. conducted in Korea should be discussed. A) “Efficacy and safety of guselkumab compared with placebo and adalimumab in Korean patients with moderate-to-severe psoriasis: post-hoc analysis from the phase III, double-blind, placebo- and active-comparator-controlled VOYAGE 1/2 trials”. J Dermatolog Treat. 2022; B) “Clinical outcomes in adult patients with plaque psoriasis treated with ustekinumab under real-world practice in Korea: A prospective, observational, multicenter, postmarketing surveillance study”. J Dermatol. 2021.
Reviewer 2 Report
An interesting paper about the use of anti il12/23, 17 and 23 in the management of psoriasis...the main problem of the paper is the small number of participants and the fact that the 3 biologics reported are not among the newest (all the drugs were respectively the first anti il12/23,17 and 23 on the market, while in the same classes are now present more effective compounds), and in literature are present confront studies with way more patients.
Still, I think the paper may be eligible to be published, after revisions:
In the introduction, a more effective clinical description of psoriasis is mandatory; such as: "Psoriasis is characterized by erythematous-squamous plaques affecting mainly extensor surfaces but spreading to all body areas . Different clinical phenotypes of psoriasis have been reported, including palmoplantar, inverse, guttate, pustular, and others." and cite: doi: 10.3390/pharmaceutics14020294. and doi: 10.3390/healthcare9050543.
Thank You
Reviewer 3 Report
To Authors
This real-world retrospective analysis provides data on the comparison of the most frequently used biologics, secukinumab, ustekinumab and guselkumab from a single center in Korea. I have several comments that the Authors should consider to improve their paper.
Minor comments
- The title should be modified to mention the 3 biologics compared and moderate-to-severe psoriasis
- Instead of using the word “major” I would suggest to re-phrase to “most frequently used” or “commonly used” when referring to the 3 biologics used in the study
- Patients had were treated with at least 1 injection of the 3 biologics with at least 56 wks follow up. There is no mention (at least that I could not see) of patients previously treated with or naïve to biologics. This is particularly important as previously biological treatment (vs native) is a recognised (negative) predictor of efficacy and/or drug survival
- The number of patients are presented in table 1 at baseline but there is no mention of the number of patients that were lost (through inefficacy or adverse events or other reasons) during follow up. The number of patients lost during follow up need to be clearly presented and discussed ( in results and/or discussion)
- Please provide median drug survival time (and range) for each biologic (time in days or weeks + 95% CI)
- “Current drinking” should be modified to “Alcohol consumption”
- Table 4 (and other tables where relevant); remove all bold text and just highlight statistically significant p values. Round down to 2 decimal places (use bold for sig p values for all tables)
- When comparing results from this study to other studies and databases such as DERMBIO and BADBIR, please state values of these other studies instead of “longer” or “similar”
Major comments
- As previously mentioned, previous biological treatment (PBT) (vs naïve) needs to be included in cox-regression models as predictor of drug survival - this information is currently absent from the baseline characteristics and if PBT is similar across the 3 groups?
- Table 1 shows baseline characteristics of patients but although most of the main variables are similar across groups, some differences are questionable. Rapidly eyeballing the data, the following variables do not appear “similar”; such as BMI (lower for UST), male (higher for SEC), PASI (higher for GUS), hypertension (lower for UST), Diabetes (lower for UST and GUS) and dyslipidemia (lower for UST) – actually, in general it would seem that patient characteristics were slightly in favour (healthier) in the UST group – further strengthening the results observed (lower PASI response and lower retention rate) – this could be mentioned too. However, it would be correct in my opinion (besides eyeballing the data) to show p-values for comparison of baseline variables for the 3 groups. This should also be extended to Table 2, even if it is less justified. Please therefore add p-values to comapre variables for the 3 biologics for tables 1 and 2 and also add PBT as a variable here too (number of patients having PBT and %).
- Please extend the limitation section to address the small sample size of patients and the short follow up period of 1 year considering that their are many other real-life studies with much longer follow up periods. Also, this study was monocentric and generalizability is limited - authors could compare their results to some other real life studies in Korea (see Kwon et al 2018 below) to see if patient charactersitics are similar and efficacy data are similar for these biologics therefore permitting therir findings ot be cautiously generalized beyond a single center
- Several other papers need to be mentioned in the context of their results in the Discussion;
Youn SW, Yu DY, Kim TY, Kim BS, Lee SC, Lee JH, Choe YB, Lee JH, Choi JH, Roh JY, Jo SJ, Lee ES, Shin MK, Lee MG, Jiang J, Lee Y. Efficacy and safety of guselkumab compared with placebo and adalimumab in Korean patients with moderate-to-severe psoriasis: post-hoc analysis from the phase III, double-blind, placebo- and active-comparator-controlled VOYAGE 1/2 trials. J Dermatolog Treat. 2022 Feb;33(1):535-541. doi: 10.1080/09546634.2020.1770174. Epub 2020 May 27. PMID: 32419536.
Reich K, Song M, Li S, Jiang J, Youn SW, Tsai TF, Choe YB, Huang YH, Gordon KB. Consistent responses with guselkumab treatment in Asian and non-Asian patients with psoriasis: An analysis from VOYAGE 1 and VOYAGE 2. J Dermatol. 2019 Dec;46(12):1141-1152. doi: 10.1111/1346-8138.15109. Epub 2019 Oct 20. PMID: 31631377.
Kwon SH, Lee ES. Secukinumab Response in Korean Patients with Moderate to Severe Plaque-Type Psoriasis Irrespective of Previous Biologic Use: 1-Year Experience at a Single Center. Ann Dermatol. 2020 Jun;32(3):255-257. doi: 10.5021/ad.2020.32.3.255. Epub 2020 Apr 24. PMID: 33911748; PMCID: PMC7992612.
Youn SW, Yu DY, Kim BS, Kim Y, Kim KJ, Choi JH, Son SW, Lee ES, Ro YS, Park YL, Lee Y, Lee JH, Park HJ, Kim TY, Lee MG, Shin MK, Choi GS, Kim DH, Jo SJ, Lee SC; Stelara PMS investigators. Clinical outcomes in adult patients with plaque psoriasis treated with ustekinumab under real-world practice in Korea: A prospective, observational, multi-center, postmarketing surveillance study. J Dermatol. 2021 Jun;48(6):778-785. doi: 10.1111/1346-8138.15670. Epub 2021 Feb 2. PMID: 33528054.
Round 2
Reviewer 1 Report
The Authors have thoroughly addressed all my comments.
Reviewer 3 Report
Thank you for fully addressing all comments. I have no additional comments on this paper.